# Vitamin B12 Ameliorates the Pathological Phenotypes of Multiple Parkinson’s Disease Models by Alleviating Oxidative Stress

**DOI:** 10.3390/antiox12010153

**Published:** 2023-01-09

**Authors:** Yue Wu, Zhongting Zhao, Naidi Yang, Chenqi Xin, Zheng Li, Jiajia Xu, Bo Ma, Kah-Leong Lim, Lin Li, Qiong Wu, Changmin Yu, Chengwu Zhang

**Affiliations:** 1Key Laboratory of Flexible Electronics (KLoFE) & Institute of Advanced Materials (IAM), School of Flexible Electronics (Future Technologies), Nanjing Tech University, Nanjing 211816, China; 2Department of Central Laboratory of Basic Medicine, The First Affiliated Hospital of Yangtze University, 8 Hangkong Road, Jingzhou 421000, China; 3Department of Diagnostic Radiology, Yong Loo Lin School of Medicine, National University of Singapore, Singapore 119074, Singapore; 4School of Pharmaceutical Sciences, Nanjing Tech University, Nanjing 210023, China; 5Lee Kong Chian School of Medicine, Nanyang Technological University, Singapore 308232, Singapore; 6The Institute of Flexible Electronics (IFE, Future Technologies), Xiamen University, Xiamen 361005, China; 7School of Basic Medical Sciences, Shanxi Medical University, 56 Xinjian Road, Taiyuan 310003, China

**Keywords:** Parkinson’s disease, vitamin B12, antioxidant, reactive oxygen species, mitochondria

## Abstract

Parkinson’s disease (PD) is the second most common neurodegenerative disease characterized by progressive loss of dopaminergic neurons in the substantia nigra of the midbrain. The etiology of PD has yet to be elucidated, and the disease remains incurable. Increasing evidence suggests that oxidative stress is the key causative factor of PD. Due to their capacity to alleviate oxidative stress, antioxidants hold great potential for the treatment of PD. Vitamins are essential organic substances for maintaining the life of organisms. Vitamin deficiency is implicated in the pathogenesis of various diseases, such as PD. In the present study, we investigated whether administration of vitamin B12 (VB12) could ameliorate PD phenotypes in vitro and in vivo. Our results showed that VB12 significantly reduced the generation of reactive oxygen species (ROS) in the rotenone-induced SH-SY5Y cellular PD model. In a Parkin gene knockout *C. elegans* PD model, VB12 mitigated motor dysfunction. Moreover, in the 1-methyl-4-phenyl-1,2,3,6-tetrahydropyridine (MPTP)-induced mouse PD model, VB12 also displayed protective effects, including the rescue of mitochondrial function, dopaminergic neuron loss, and movement disorder. In summary, our results suggest that vitamin supplementation may be a novel method for the intervention of PD, which is safer and more feasible than chemical drug treatment.

## 1. Introduction

Parkinson’s disease (PD) is the second most common neurodegenerative disease and is characterized by the loss of dopamine (DA) neurons in the substantia nigra (SN) of the midbrain [1,2]. Clinically, PD patients exhibit motor and nonmotor system problems [3,4]. The degeneration of DA neurons seems to specifically occur in the SN since those in the apposed ventral tegmental area are largely unaffected [5]. The death of DA neurons in the SN leads to the depletion of striatal DA, which in turn results in a series of movement disorders [6,7]. The etiology of PD is thought to involve multiple factors, including genetics, the environment, and aging [8,9,10]. However, the underlying mechanism of neurodegeneration in DA neurons remains to be elucidated, but oxidative stress and mitochondrial dysfunction are strongly implicated by many studies [8,11].

Reactive oxygen species (ROS) are an essential component of cellular homeostasis and are usually produced during mitochondrial electron transfer chains (ETCs) or redox reactions [12]. Normally, ROS levels in the body are maintained homeostatically, but internal or external stimuli may disrupt the balance and cause disease. For example, rotenone (Rot), a mitochondria complex I inhibitor, can induce excessive ROS generation and lead to severe oxidative stress, which is a causative factor of PD [13,14]. Rot induces liquefaction necrosis and selective DA cell degeneration in the midbrain [15,16]. 1-Methyl-4-phenyl-1,2,3,6-tetrafluoropyridine (MPTP) can also cause oxidative stress and is commonly used as an inducer of PD in experimental models, especially in vivo [17]. When oxidized by monoamine oxidase B, MPTP is metabolized to MPP^+^, which enters DA in the substantia nigra compact (SNc) to produce neurons through the DA reuptake system [18]. After entering cells, MPP^+^ inhibits the mitochondrial ETC complex I enzyme NADH-ubiquinone oxidoreductase, resulting in mitochondrial electron leakage and ROS production [19].

Although PD remains incurable, clinical intervention could alleviate the symptoms of patients. Of the treatments, vitamin administration shows promising effects [20,21,22,23]. Some vitamins, such as vitamin C and vitamin E, possess antioxidant ability. High concentrations of vitamin C can reduce oxidation free radicals and quinones derived from DA by quenching O_2_^•−^ and DA^•−^ [24]. Vitamin E could inhibit the activity of 15-lipoxygenase via reduction of the enzyme’s nonheme iron from its active Fe^3+^ to an inactive Fe^2+^ state, upregulating the antioxidant transcription factor nuclear factor 2-related factor 2 (Nrf2) [25,26]. In addition to its well-known antioxidant activities, vitamin E also exerts anti-inflammatory effects and prevents DA neuron apoptosis [27]. Moreover, vitamin D is found to enhance the release of antioxidant substances such as glutathione (GSH) and superoxide dismutase (SOD) and displays therapeutic effects on PD [28].

Notably, in PD patients, the vitamin B12 (VB12) level is found to be significantly downregulated in tissues including blood, which has been utilized for the diagnosis of PD [29,30,31]. Dietary VB12 supplementation in PD patients reduces homocysteine levels and alleviates the manifestations of PD in patients [32,33]. In addition, VB12 is reported to inhibit the β-sheet conformation formation of α-synuclein (α-SN) in a dose-dependent manner and reduce the cytotoxicity of α-SN aggregates [34]. 5′-deoxyadenosinecoamin (AdoCbl), a physiological form of VB12, inhibits LRRK2 kinase activity by disrupting LRRK2 dimerization and reduces its neurotoxicity in cultured primary neurons and Drosophila PD models [35]. Thus far, whether VB12 serves as an antioxidant to protect against PD has not been illustrated.

In the present study, multiple oxidative stress-related PD models were subjected to VB12 treatment, and the rescue effects and molecular mechanisms were investigated. The results showed that VB12 could ameliorate PD phenotypes in vitro and in vivo. Thus, VB12 holds great potential for the treatment of PD.

## 2. Materials and Methods

### 2.1. Materials

All reagents and solutions were purchased from the supplier used without any further purification. The medium for the cell culture was from Thermo Fisher Scientific (Waltham, MA, USA). Fetal bovine serum (FBS) was purchased from Bioind Biological Industry (Kibbutz Beit Haemek, Israel). VB12 (red powder, dissolved in distilled water, purity > 98%) was obtained from Biofrox (Einhausen, Germany). Rot was purchased from Sigma–Aldrich (St. Louis, MO, USA). MPTP was purchased from Beyotime Biotechnology Institute (Shanghai, China). All detection kits (3-(4,5-dimethylthiazol-2-yl)-2,5-diphenyl-tetrazolium bromide (MTT), propidium iodide (PI), triphenyladenine (ATP), ROS, SOD, and GSH) were purchased from Beyotime Biotechnology Institute (Shanghai, China). Peroxisome proliferator-activated receptor-γ coactivator (PGC-1α), c-Jun N-terminal kinase (JNK), and phosphorylated (p)-JNK were obtained from Cell Signaling Technology (Danvers, MA, USA). Glyceraldehyde-3-phosphate dehydrogenase (GAPDH) was purchased from Sigma–Aldrich (St. Louis, MO, USA). Anti-mouse and anti-rabbit secondary antibodies were purchased from Immunoway Biotechnology (Plano, TX, USA). Fluorescence imaging experiments were recorded on a Zeiss LSM880 NLO (2 + 1 with BIG) confocal microscope.

### 2.2. Cell Culture and VB12 Treatments

Human SH-SY5Y neuroblastoma cells (provided by National Neuroscience Institute of Singapore) were cultured with Dulbecco’s modified Eagle’s medium (DMEM) (containing 10% FBS and 1% penicillin/streptomycin) and inoculated into culture dishes in a 5% CO_2_ incubator at 37 °C. VB12 stock solution (9 mM/L sterile water, filtered by 0.22 µm pore size) was prepared immediately prior to use and diluted with cell culture medium to the desired final concentration.

### 2.3. Cell Viability Assays

Cell viability was determined by MTT assay [36]. After cell treatment, (5 mg/mL) was added to each well to reach a final concentration of 0.5 mg/mL and cultured for another 3–4 h. Then, the medium containing MTT was removed, and dimethylsulfoxide (DMSO) was added to dissolve formazan. After standing for 10 min, the absorbance at 490 nm was measured, and cell viability was normalized to the percentage of the control group.

### 2.4. Intracellular ROS Content, PI, SOD, GSH and ATP Assays

SH-SY5Y cells were inoculated into the culture dish and cultured until the confluence was approximately 60–70%. The cells were pretreated with 1.5 μM VB12 for 3 h and then cocultured with 1 μM Rot for 24 h [37]. Subsequent experiments were performed.

ROS Assays: We used the DCFH-DA probe to detect ROS [38]. Each confocal dish was added to 500 μL DMEM containing 10 μM DCFH-DA and cultured for 20 min. After that, DCFH-DA was completely removed with 3 washes of PBS. Finally, the cell ROS level was evaluated under 488 nm excitation and 525 nm emission.

PI Assays: The pretreatment of cells was the same as described above. Then, the medium was removed, and 1 mL PI working solution (C2015S) was added to detect fluorescence after 30 min of incubation (PI E_x_/E_m_ = 535/617 nm) [39].

SOD Assays: Cells were collected and washed 1–2 times with precooled PBS or normal saline at 4 °C or in an ice bath. Precipitation was homogenized with precooled PBS at 4 °C or in an ice bath (glass homogenizers or various common electric homogenizers can be used). Then, the homogenate was centrifuged at 4 °C, and the supernatant was taken as the sample to be tested for detection according to the SOD instructions [40].

GSH Assays: The cells were washed with PBS and collected by centrifugation, and the supernatant was aspirated. Protein removal reagent M solution was added at 3 times the volume of the cell precipitate. Then, the samples were freeze–thawed twice using liquid nitrogen and a 37 °C water bath; following that, 4 °C or ice bath for 5 min. Centrifugation was performed at 10,000× *g* at 4 °C for 10 min. The supernatant was taken for GSH determination according to the instructions [41].

ATP Assays: The cells were inoculated in six-well plates, and VB12 and Rot were treated in the same way as described above. After 24 h, 200 μL of ATP lysis solution was added to each well according to the instructions, and the ATP content was detected. At the same time, the total protein concentration was detected, and the ATP content per mg protein was compared [42].

### 2.5. Western Blotting

Cells were inoculated and treated as above, and protein concentrations were assessed by bicinchoninic acid (BCA) assay (Beyotime Shanghai, China). The proteins were separated in equal quantities (20 µg) on 10% or 8% gels by sodium dodecyl sulfate–polyacrylamide gel electrophoresis and transferred to nitrocellulose membranes by electrophoresis (Sigma–Aldrich, St. Louis, MO, USA). Subsequently, western blotting was performed using anti-PGC-1α, p-JNK, JNK or anti-GAPDH antibodies. GAPDH was used as a loading control for cell lysates to normalize PGC-1α and p-JNK levels. Finally, a BeyoECL Plus Ultra-sensitive ECL Chemiluminescence Kit (Beyotime Shanghai, China) and Omega Lum W imaging system BioRad ChemiDoc XRS imaging system (Hercules, CA, USA) were used to observe the immune response bands corresponding to PGC-1α and p-JNK.

### 2.6. Effects of VB12 on the Behavior of C. elegans PD Model

All *C. elegans* (N2 and VC1024) were obtained from the Caenorhabditis Genetics Center (University of Minnesota, MN, USA). N2 (wild-type *C. elegans*) and VC1024 (parkin-null *C. elegans*) were maintained and reproduced at 20 °C in *C. elegans* growth medium (NGM) with E. coli uracil strain OP50 seeds as a nutriment. To eliminate any effects caused by the difference between *C. elegans* ages, each experiment was carried out in the synchronous *C. elegans* population. N2 and VC1024 were cultured in common NGM medium, and the experimental group was cultured in medium containing 50 μM and 100 μM VB12, respectively. Imaging was performed using L4 *C. elegans*. Each group of *C. elegans* was immersed in an aqueous solution containing an ROS probe (DCFH-DA) for 5 min, then crawled on an OP50-free NGM for 30 min, and finally transferred to a slide coated with 2% agarose gel. Imaging was carried out with a Zeiss LSM880 NLO (2 + 1 with BIG). The study on the movement distance of *C. elegans* was selected as a behavioral experiment. At least 30 *C. elegans* were removed from each group, and the movement distance of each *C. elegans* in OP50 free medium in 1 min was calculated. In this study, the movement distance of *C. elegans* was calculated by the number of complete sinusoidal movements.

### 2.7. Effect of VB12 on the Mouse PD Model

C57BL/6 mice were purchased from Qinglongshan Animal Breeding Farm, and the handling and treatment of animals were approved by the Institutional Animal Care and Use Committee (IACUC) of Nanjing Tech University. The mice were kept in a light-dark cycle for 12 h. Mice (4 weeks of age, 20 g, male) were divided into 4 groups (6 mice for each group): sham control, MPTP-induced PD model, VB12-treated PD model, and VB12 only. Sham control mice were not subjected to any treatment; PD model mice were subjected to intraperitoneal injection of MPTP dissolved in saline for 10 days (20 mg/kg/d); VB12-treated PD model were subjected to oral gavage VB12 (250 mg/kg/d) one week before MPTP injection and continued for 20 days; and VB12 only mice received no MPTP injection. Behavioral analysis, including the open-air test and pole test, was performed 20 days after PD model establishment. The open-air test recorded the distance traversed by the mice while standing upright in the box with 5 × 5 (10 cm side length) squares at the bottom [43]. The pole test recorded the time for the mice to climb from the top to the bottom of a 50 cm long rod [44]. After that, the mice were perfused with saline and 4% paraformaldehyde. The brain tissue was collected and subjected to tyrosine hydroxylase immunohistochemistry staining. The scanned photos were captured by CASE viewer and analyzed using Imaje J.

### 2.8. Statistical Analysis

The experiments were repeated at least three times, and all values are expressed as the mean ± SEM. Statistical differences were analyzed by one-way ANOVA. Differences at the 95% level were considered significant (*p* < 0.05).

## 3. Results

### 3.1. Effect of VB12 on PD Cell Viability

Rot, a mitochondrial complex inhibitor, is often used to establish cellular PD models. Referring to the literature and our previous study [45], 1 μM Rot was selected to create the SH-SY5Y PD model (Figure 1A). Before application to the cellular PD model, the cytotoxicity of VB12 was assessed. As shown in Figure 1B, VB12 at lower concentrations had negligible cell toxicity, and cell viability was 80% even with VB12 at 100 μM. Pretreatment with VB12 revealed that the optimal dose is 1.5 μM in protecting these cells against Rot-induced cell death. At higher concentrations, VB12 did not show the same effect (Figure 1C). To further confirm the protective effect of VB12, PI fluorescence imaging was performed. As shown in Figure 1D,E, 1.5 μM VB12 significantly reduced Rot-induced cell death, as illustrated by PI staining. These results demonstrate that VB12 could rescue Rot-induced SH-SY5Y cell death.

### 3.2. VB12 Relieves Oxidative Stress in the Rot-Induced Cellular PD Model

Given that Rot inhibits mitochondrial complex I activity and induces oxidative stress, we checked whether VB12 achieves its rescue effect in a cellular PD model by alleviating oxidative stress. Rot treatment induced ROS generation in SH-SY5Y cells, as shown by the ROS fluorescence probe DCFH-DA, which was significantly reduced by VB12 administration (Figure 2A,B). To further verify the antioxidant activity of VB12, we treated the cellular PD model with the ROS scavenger N-acetylcysteine (NAC), which rendered an effect comparable to that of VB12 (Figure 2A,B). Because Rot treatment inhibited complex I activity and reduced ATP production, we checked whether VB12 administration could mitigate this effect in SH-SY5Y cells. As shown in Figure 2C, VB12 administration restored the ATP reduction induced by Rot.

Cells maintain their redox homeostasis by regulating ROS and reductant levels. Therefore, we further examined whether VB12 treatment would also affect the reductants of cells. We determined the level of GSH and SOD in Rot-treated SH-SY5Y cells with a specific kit. Rot reduced GHS and SOD levels, whereas VB12 significantly restored these levels. Compared with the control, VB12 only did not affect GSH or SOD levels (Figure 3A,B).

### 3.3. VB12 Affects Oxidative-Related Proteins in Rot-Treated SH-SY5Y Cells

After confirming the antioxidant effect of VB12, we wanted to determine the possible underlying molecular mechanism. JNK is the key kinase activated by oxidative stress, and its activation leads to cell death [46]. We found that Rot treatment induced JNK phosphorylation, whereas VB12 reduced the p-JNK level (Figure 4A,B). PGC-1α, a transcriptional coactivator that activates mitochondrial biogenesis, is often strongly implicated in oxidative stress [47]. As shown in Figure 4A,B, Rot treatment reduced the PGC-1α expression level, whereas VB12 reversed it (Figure 4A,C).

### 3.4. VB12 Reduces ROS and Ameliorates Movement Deficiency in the C. elegans PD Model

Given the promising neuroprotective effects of VB12 in vitro, we further explored its application and effect in vivo. *C. elegans* has a well-characterized nervous system and shares the same neurotransmitters (e.g., DA). Therefore, *C. elegans* are amendable to model neurological diseases such as PD [48]. VC1024 is a strain of *C. elegans* without the parkin gene that recapitulates the salient features of PD, including motor deficits. We found that VC1024 *C. elegans* had a higher level of ROS, as seen by the ROS fluorescence probe (DCFH-DA), than the N2 control counterpart. VB12 treatment significantly reduced the intensity of the ROS fluorescence signal in these mutant worms, whereas VB12 itself did not induce a detectable change in ROS in control *C. elegans* (Figure 2A,B). As VC1024 *C. elegans* displayed movement deficits, we checked whether VB12 treatment could ameliorate these deficits. VB12 treatment alleviated the bradykinesia of VC1024, as recorded by the body bends of *C. elegans* (Figure 5C).

### 3.5. VB12 Ameliorates Movement Deficits and DA Neuron Loss in the MPTP-Induced PD Model

Having demonstrated the antioxidant effect of VB12 in a cell line and *C. elegans* model of PD, we wished to determine whether VB12 also exerts comparable efficacy in a mammalian PD model. We established a classic PD mouse model by intraperitoneal injection of MPTP, a neurotoxin often adopted to induce PD symptoms in vivo [49]. One week before MPTP injection, VB12 was administered by oral gavage and continued until the behavioral test 20 days after model establishment. As shown in Figure 5A–C, VB12 administration significantly improved the movement deficit of PD mice, as revealed by the open field (Figure 6A,B) and pole tests (Figure 6C). To further check if the DA neuron number was affected, TH immunostaining of the SN was carried out. MPTP reduced the number of TH-positive neurons in the SNc, whereas VB12 administration ameliorated DA neuron loss (Figure 6D,E).

## 4. Discussion

In this study, we found that VB12 treatment could ameliorate Rot-induced oxidative stress and apoptosis in SH-SY5Y cells. In vivo, VB12 reduced endogenous ROS and improved the motorability of the *C. elegans* PD model. In the MPTP-induced mouse PD model, VB12 treatment alleviated movement deficits and degeneration of DA neurons in the midbrain.

PD, as the second most common neurodegenerative disease after Alzheimer’s disease, is characterized by DA neuron loss in the SN of the midbrain and is accompanied by motor and nonmotor manifestations [50,51]. Oxidative stress is thought to be one of the pivotal causative factors of PD. Mechanistically, ROS lead to the oxidation of biomacromolecules such as DNA, lipids, and proteins, which destroys their normal function and eventually results in the pathogenesis of PD [52,53]. Moreover, ROS can induce mitochondrial dysfunction, which is crucial for DA neurons of the SNc since they require more ATP to maintain resting membrane potential, propagate action potentials, and achieve synaptic transmission [9,54]. Reducing ROS has always been a potential treatment for PD [55,56]. In the present study, we investigated the antioxidant effect of VB12 in multiple PD models.

In the cellular PD model, we found that VB12 reduced Rot-induced ROS elevation, increased ATP production, and increased the levels of antioxidants, including GSH and SOD. Redox homeostasis is crucial for cells. Once it is disrupted, diseases such as PD can occur [50]. Previously, Tian Wang and colleagues reported that ROS levels were elevated in a PD cellular model [57]. Supplementation with GSH or SOD rescued neuronal death in cell and mouse PD models [58,59]. These results were consistent with what was observed in our VB12-treated SH-SY5Y cells. Oxidative stress can induce mitochondrial dysfunction and ATP depletion [60]. We observed that VB12 significantly increased the expression of PGC-1α protein and ATP production in the Rot-induced SH-SY5Y PD model. JNK is a mitogen-activated protein kinase that is closely related to oxidative stress [61,62]. Phosphorylation of JNK has been implicated in various PD models [63,64]. In the present work, we showed that VB12 reduces p-JNK levels and alleviates Rot-induced cell death. These results suggest that VB12 could ameliorate oxidative stress and rescue ROS-induced cell death by regulating the ROS signaling pathway.

*C. elegans* is a widely adopted model in PD research due to its relatively simple nervous system and transparent body [65]. He et al. reported an increase in ROS and movement disorder in a *C. elegans* model of PD [66]. Substances with neuroprotective or antioxidant properties reduced ROS and alleviated dyskinesia in *C. elegans* [67,68]. We found that VB12 treatment reduced ROS levels in parkin-null *C. elegans* VC1024. In addition, VB12 also improved its movement deficit. This suggested that VB12 could not only protect against oxidative stress in vitro but also displayed antioxidant activity in vivo.

We also investigated whether VB12 exerted comparable effects in a mammalian PD model. As expected, in the MPTP-induced mouse PD model, VB12 rescued DA neuron death in the SN and alleviated movement deficits. Rui et al. reported that baicalein reduced the damage to DA neurons in PD mice and improved behavioral ability [69]. Many studies have found that preventing the reduction in SN in PD mice can effectively relieve the symptoms of PD [50,51,52,53,54,55,56,57,58,59,60,61,62,63,64,65,66,67,68,69,70,71,72]. Our results provide evidence supporting that oral administration of VB12 could potentially serve as an effective intervention for the treatment of PD in the clinic.

The role of nutritional factors such as VB12 in the human body is relatively complex. It is the cofactor of two enzymes, namely methylmalonyl coenzyme a synthase and methionine synthase, and is crucial to the biosynthesis of methionine and nucleotides. VB12 deficiency leads to the accumulation of metabolic substrates (homocysteine), which changes immune homeostasis and leads to atherosclerotic diseases, including ischemic stroke [73]. VB12 supplementation in the diet of PD patients could reduce the level of homocysteine in PD patients and alleviate the symptoms of PD patients [34,74]. Previous literature has explored the relationship between VB12 and PD [32,33,34]. However, the pathogenesis of PD is not yet clear. As oxidative stress is considered one of the key pathogenic factors of PD, we aimed to explore the relationship between VB12 and PD from the perspective of oxidative stress. Mechanistically, our results prove that VB12 increased the levels of SOD and GSH, reduced the generation of ROS, and enhanced ATP levels. Moreover, our results also indicated that VB12 displayed antioxidant activity in vivo. However, the underlying mechanisms of how VB12 affects the oxidative-related proteins in Rot-treated SH-SY5Y cells and whether VB12 interacts with Rot or its metabolites to inhibit Rot-induced ROS production need further investigation. In addition, there are no clinical trials of VB12’s therapeutic effects on humans yet, and there is still a way to go in human PD patients. Nevertheless, our results provide evidence that VB12 could potentially serve as an effective intervention for the treatment of PD in the clinic.

## 5. Conclusions

In conclusion, we systematically investigated the antioxidant activity of VB12 in multiple PD models. We demonstrated that VB12 significantly ameliorated oxidative stress by reducing ROS and alleviating reductants in a cellular PD model. In vivo, VB12 rescued the movement deficit of *C. elegans* and the mouse PD model. Mechanistically, VB12 reduced JNK phosphorylation and improved the mitochondrial biogenesis regulator PGC-1α (Figure 7). Given its easier availability and better biocompatibility, VB12 holds great potential to serve as an alternative intervention for the treatment of PD.

## Figures and Tables

**Figure 1 antioxidants-12-00153-f001:**
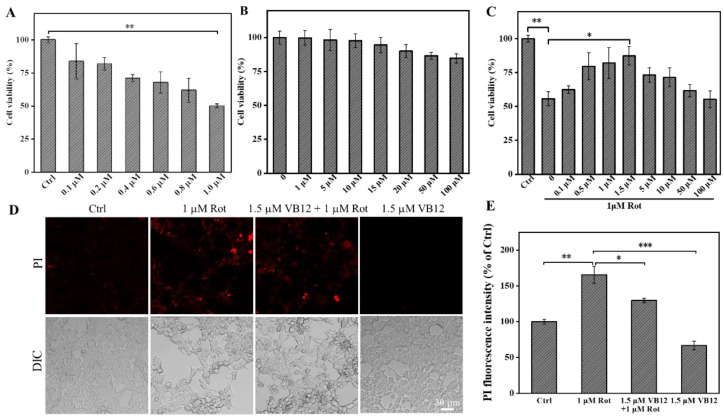
VB12 rescued Rot-induced SH-SY5Y cell death. (**A**) Survival rate of cells under different concentrations of Rot (0.1, 0.2, 0.4, 0.6, 0.8, and 1 μM). (**B**) The cytotoxicity of VB12 at different concentrations. (**C**) Cell viability of SH-SY5Y cells pretreated with VB12 when exposed to Rot. (**D**) SH-SY5Y cells were pretreated with VB12 at the specified concentration for 3 h and exposed to Rot (1 μM) for an additional 24 h. (**E**) Quantization of PI fluorescence. Scale bar = 30 μm. Data are the mean ± SEM, n = 3. * *p* < 0.05, ** *p* < 0.01, *** *p* < 0.001 compared with other groups.

**Figure 2 antioxidants-12-00153-f002:**
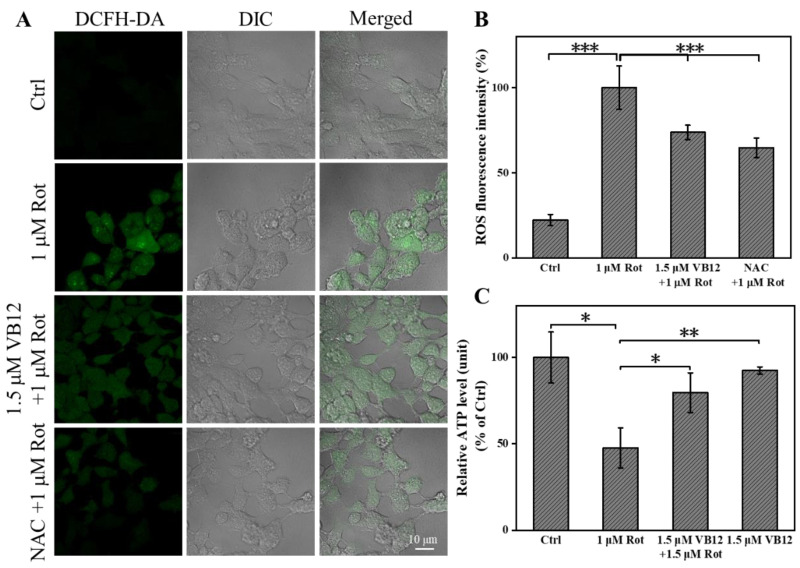
VB12 decreases intracellular ROS and increases ATP content. (**A**) Fluorescence imaging of intracellular ROS treated with Rot, Rot+VB12, or Rot+NAC. (**B**) Fluorescence intensity quantification of (A). (**C**) ATP content analysis of SH-SY5Y cells. Scale bar = 10 μm. Data are the mean ± SEM, n = 3. * *p* < 0.05, ** *p* < 0.01, *** *p* < 0.001.

**Figure 3 antioxidants-12-00153-f003:**
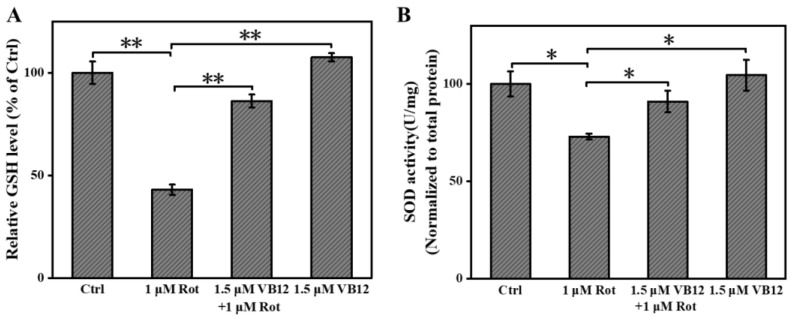
VB12 restored the reductant level of Rot-treated SH-SY5Y cells. (**A**) GSH content. (**B**) SOD content. Data are the mean ± SEM, n = 3. * *p* < 0.05, ** *p* < 0.01 compared with other groups.

**Figure 4 antioxidants-12-00153-f004:**
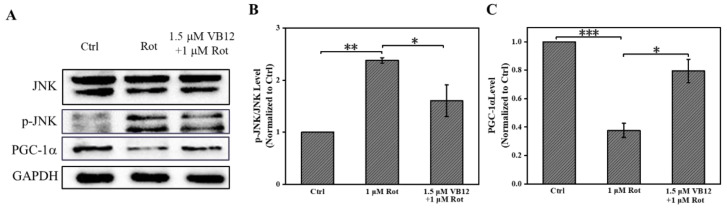
VB12 mediates oxidative stress-related proteins. (**A**) Relative protein levels of JNK and PGC-1α as shown in the Western blot. (**B**) Quantification of the relative protein level of p-JNK. (**C**) Quantification of the relative protein level of PGC-1α. Data are the mean ± SEM, n = 3. * *p* < 0.05, ** *p* < 0.01, *** *p* < 0.001.

**Figure 5 antioxidants-12-00153-f005:**
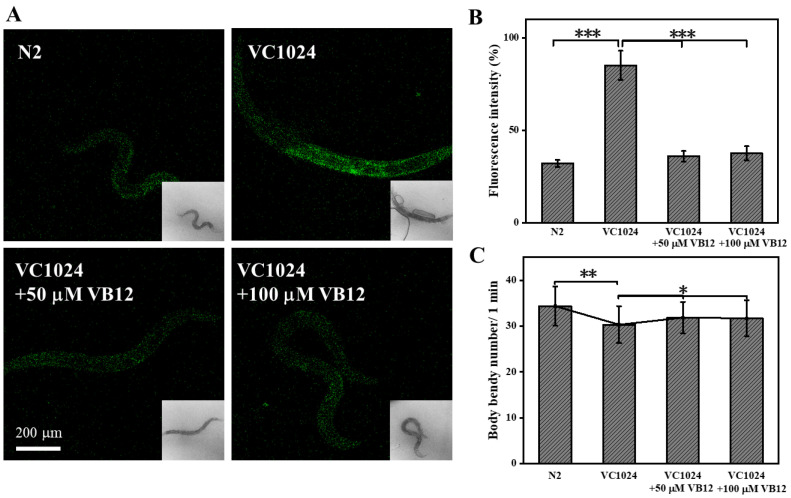
VB12 relieves symptoms of PD *C. elegans*. (**A**) Fluorescent intensity of ROS in wild-type (N2) and PD *C. elegans* (VC1024) with and without VB12 treatment; (**B**) quantification of the fluorescence intensity of the ROS probe shown in (**A**); (**C**) the number of sinusoidal torsion in *C. elegans* of wild-type and PD with/without VB12 treatment. Scale bar = 200 μm. Data are the mean ± SEM, n = 30. * *p* < 0.05, ** *p* < 0.01, *** *p* < 0.001 compared with other groups.

**Figure 6 antioxidants-12-00153-f006:**
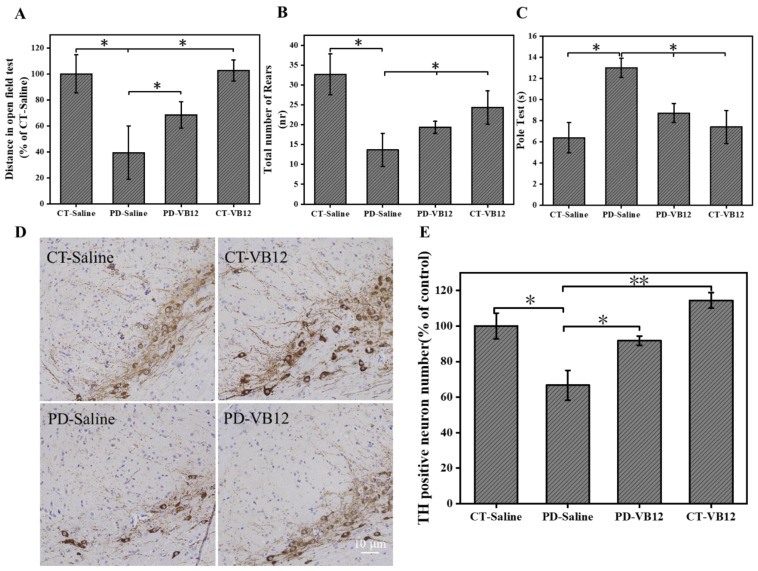
VB12 alleviated behavioral deficits and DA neuron loss in PD mice. (**A**) Open field assay to show the ambulation distance and (**B**) rearing frequency of mice; (**C**) pole test to show the movement coordination of mice; (**D**) representative TH immunohistochemistry staining of DA neurons of SN neurons of PD; (**E**) quantification of TH neurons number by ImageJ. Scale bar = 10 μm. Data are the mean ± SEM, n = 4. * *p* < 0.05, ** *p* < 0.01 compared with other groups.

**Figure 7 antioxidants-12-00153-f007:**
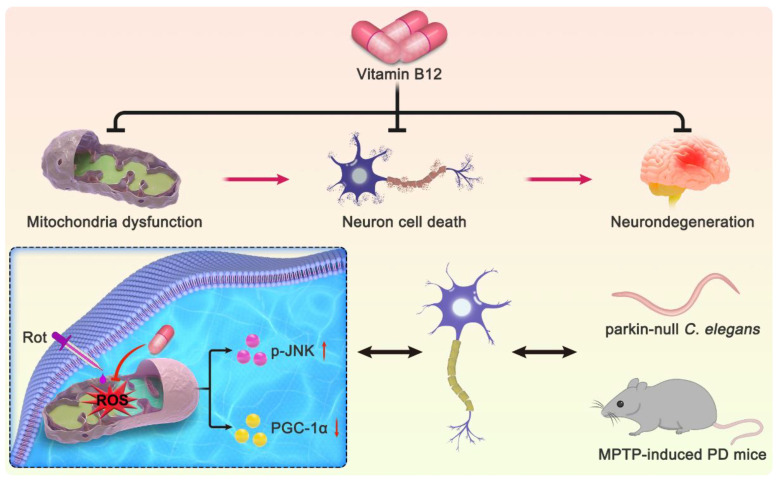
VB12 effectively alleviated the symptoms of PD models in vitro and in vivo.

## Data Availability

The data are contained within the article.

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
