# Peer review of "Vitamin B12 Ameliorates the Pathological Phenotypes of Multiple Parkinson’s Disease Models by Alleviating Oxidative Stress"

_antioxidants, 2023, doi:10.3390/antiox12010153_

Round 1

Reviewer 1 Report

Congratulations to the authors!  Modeling of Vitamin B12 effects has been a knowledge gap for a long time. My minor comment is that all the bar graphs must indicate results of ANOVA vs T-tests, including multitest results.

In addition, the discussion needs to include a paragraph of limitation. For example, the authors are unable to specify whether VIB12 acts on rotenone itself or it's metabolites; potential difference between human and experimental systems; and no clinical trials have verified VIB12's therapeutic effects yet.

A recent meta-analysis (PMID: 31733593) should be cited in the introduction. 

Author Response

Thanks for your constructive comments on our mamuscript.  Please refer to the attachment for our point-by-point response.

Reviewer 2 Report

The manuscript of Yue et al. describes the antioxidant effects of VB12 in several Parkinson’s disease model. After repairing these missing details and inaccuracies, the manuscript can be considered for publication.

1.   Method 2.4, it will be more clearly to separate those assays. 

2.   Method 2.7, no detail methods about behavioral tests. 

3.  In Fig.1D and E, I can’t tell any signal in 1.5uM VB12 of PI staining, could you explain why the quantification shows around 40% reduce than CTL group?

4.  TH staining result in Fig.6D, the number of TH positive neuron in CT-VB12 group showed in figure and quantification is more than CT-saline, is there significant different between those 2 group? How to explain this phenomenon? Need to choose the appropriate figures.

5.   Is there more evidence can prove the VB12’s neuroprotection in MPTP-PD model?  Like, the TH protein expression in midbrain and the TH or DAT expression in striatum?

6.   Is there any evidence can prove the antioxidant function in C.elegans or MPTP mice model?

Author Response

Thanks for your critical and constructive comments on our manuscript. Please refer to the attachment for the point-by-point response.

Reviewer 3 Report

General comment:

This study aims to identify whether administration of vitamin B12 can ameliorate the Parkinson’s disease (PD) phenotypes in rotenone-31 induced SH-SY5Y cellular PD model, Parkin gene knockout C. elegans PD model and MPTP induced mouse PD model.

The authors found that vitamin B12 treatment could ameliorate Rot induced oxidative stress and apoptosis of SH-SY5Y. In vivo, vitamin B12 decreased the endogenous ROS and improved the motor function in the C. elegans PD model. In the MPTP-induced mice PD model, VB12 treatment alleviated movement deficit and degeneration of DA neurons. Generally speaking, this research proposal is interesting since this research could contribute to developing the potential and feasible treatment of PD.

However, some of the methodologies are not mentioned clearly in the current manuscript. The following few additional points may require further revisions.

Specific comments:

1. In the 2. Materials and Methods. 2.1-2.7 part. The authors mentioned the materials and procedures for the experiment in vitro and in vivo. However, less of references from previous studies or articles were well-cited. Based on the suitable and detailed citations, it can help researchers easily follow the detailed procedures.

 2. The Statistical Analysis should be confirmed for the post-hoc analysis.

The authors only mentioned that the statistical differences were analyzed by one-way ANOVA and T test. However, for the post-hoc analysis, the T-test is not suitable for more than 3 groups. To prevent the type I error, the authors should address the correct post-hoc analysis followed by one-way ANOVA.

3. The behavioral tests, including the open-filed test and pole test in line 180. The authors should address the procedures or the meanings of the parameters more clearly and in detail. Especially which parameters (distance, number of rear and pole test in time duration) were recorded and how to do the analysis.

4. In Figure 6 D. the representative TH immunohistochemistry staining of DA neurons of SN neurons of PD. The areas selected in the SN of four mice are not matched. (Fig 6E). The pattern is also not matched with the results (Fig 6E). Fig 6E showed the % of control. However, the number of DA neurons between CT-Saline and PD-saline is not very matched with the average results.

5. It is too short and brief in the discussion. It is suggested that the authors may address and explain more possible mechanisms and related research to compare or support the findings in the current study.

Author Response

Thanks for your critical and constructive comments on our mauscript. Please refer to the attachment for point-by-piont response

Round 2

Reviewer 1 Report

comments addressed

Reviewer 2 Report

The authors have addressed the comments, and I would suggest the acceptance. 

Reviewer 3 Report

The manuscript has been well-revised followed by the reviewer's comments.